# Role of a National Health Service Electronic Prescriptions Database in the Detection of Prescribing and Dispensing Issues and Adherence Evaluation of Direct Oral Anticoagulants

**DOI:** 10.3390/healthcare12100975

**Published:** 2024-05-09

**Authors:** Anna Gavrilova, Maksims Zolovs, Dins Šmits, Anastasija Ņikitina, Gustavs Latkovskis, Inga Urtāne

**Affiliations:** 1Department of Pharmaceutical Chemistry, Faculty of Pharmacy, Rīga Stradiņš University, LV-1007 Riga, Latvia; 2Statistical Unit, Faculty of Medicine, Rīga Stradiņš University, LV-1007 Riga, Latvia; 3Institute of Life Sciences and Technology, Daugavpils University, LV-5401 Daugavpils, Latvia; 4Department of Public Health and Epidemiology, Faculty of Health and Sports Sciences, Rīga Stradiņš University, LV-1007 Riga, Latvia; 5Faculty of Medicine, Rīga Stradiņš University, LV-1007 Riga, Latvia; 6Institute of Cardiology and Regenerative Medicine, University of Latvia, LV-1586 Riga, Latvia; 7Latvian Center of Cardiology, Pauls Stradins Clinical University Hospital, LV-1002 Riga, Latvia

**Keywords:** medication error, atrial fibrillation, rivaroxaban, dabigatran, edoxaban, apixaban, PDC, utilization, medicine, direct oral anticoagulants

## Abstract

Background: Anticoagulation therapy plays a crucial role in the management of atrial fibrillation (AF) by significantly reducing the risk of stroke. Direct oral anticoagulants (DOAC) became preferred over warfarin due to their superior safety and efficacy profile. Assessing adherence to anticoagulation therapy is necessary in clinical practice for optimising patient outcomes and treatment efficacy, thus emphasising its significance. Methods: A retrospective study utilised the Latvian National Health Service reimbursement prescriptions database, covering prescriptions for AF and flutter from January 2012 to December 2022. The proportion of days covered method was selected for adherence assessment, categorising it into three groups: (1) below 80%, (2) between 80% and 90%, and (3) above 90%. Results: A total of 1,646,648 prescriptions were analysed. Dabigatran prescriptions started declining after 2020, coinciding with a decrease in warfarin prescriptions since 2018. The total adherence levels to DOAC therapy were 69.4%. Only 44.2% of users achieved an adherence level exceeding 80%. The rate of paper prescriptions decreased from 98.5% in 2017 to 1.3% in 2022. Additionally, the utilisation of international non-proprietary names reached 79.7% in 2022. Specifically, 16.7% of patients selected a single pharmacy, whereas 27.7% visited one or two pharmacies. Meanwhile, other patients obtained medicines from multiple pharmacies. Conclusions: The total adherence level to DOAC therapy is evaluated as low and there was no significant difference in age, gender, or “switcher” status among adherence groups. Physicians’ prescribing habits have changed over a decade.

## 1. Introduction

Atrial fibrillation (AF) is a significant cardiac rhythm disorder that has become a major concern in the 21st century, with its prevalence increasing over time. AF is highly prevalent with a lifetime risk of about 1 in 3–5 individuals after the age of 45 years [1,2]. The condition is associated with increased morbidity and mortality, leading to a high burden on the healthcare system [2]. Wealthier countries have higher mortality rates compared to other countries in the EU and men are more likely to be affected than women [3].

Anticoagulation therapy plays a crucial role in the management of AF by significantly reducing the risk of stroke in individuals with this condition [2]. The decision to use anticoagulation therapy for patients with AF depends on several factors, including the individual’s risk of thromboembolism and bleeding complications [4]. Over the last decade, direct oral anticoagulants (DOAC) like rivaroxaban, dabigatran, edoxaban, and apixaban have become popular options for stroke prevention in patients with non-valvular AF, as an alternative to warfarin [5,6,7,8,9]. DOACs are preferred over warfarin due to their superior safety and efficacy profile, which includes a lower risk of serious bleeding events and stroke. They also offer the convenience of fixed dosing and eliminate the need for routine blood monitoring, simplifying patient management [10,11].

Non-adherence to warfarin therapy has long been recognised in clinical practice, with factors such as the burdensome requirement for frequent monitoring, dietary restrictions, and the high incidence of drug interactions cited as significant contributors to this phenomenon [6,12]. However, non-adherence to DOAC therapy in patients with AF also can have significant implications. Patients who do not follow their prescribed DOAC treatment regimen may not receive the full benefit of the therapy, resulting in a lower profit compared to what is observed in controlled trials [13]. Assessing adherence to anticoagulation therapy is necessary in clinical practice for optimising patient outcomes and treatment efficacy, thus emphasising its significance [14].

Through the increasing adoption of DOACs for AF management, it is important to research their real-world utilisation patterns. Medication utilisation outcomes such as adherence, persistence, discontinuation, and switching rates are crucial surrogate markers that reflect patient preferences, prescriber behaviour, and pharmacy practices [15,16,17,18,19]. These markers can be valuable not only for clinical practice but also for government decision making, as well as for developing new programmes or changes in regulation. The European Medicine Agency defines a medication error as ‘an unintended failure in the drug treatment process that leads to, or has the potential to lead to, harm to the patient’. It has been reported that the most common preventable causes of adverse events and significant public health burdens are errors in prescribing, dispensing, storing, preparing, and administering medication [20]. However, reimbursement prescription databases are not extensively utilized in Latvia to quantitatively identify prescription-related errors and non-adherence.

The aim of this study was to analyse prescribing and dispensing patterns, including evaluation of DOAC therapy adherence, among patients with AF based on data from the National Health Service’s (NHS) reimbursement prescriptions database.

## 2. Materials and Methods

### 2.1. Study Subjects and Data Collection

To examine the utilisation patterns of DOAC and other medicine within the context of AF management, a retrospective study was performed using the prescription dataset maintained by the Latvian NHS during the timeframe spanning from January 2012 to December 2022. The investigation encompassed prescriptions associated with the International Statistical Classification of Diseases and Related Health Problems (ICD) coded diagnoses pertaining specifically to AF and flutter (I48.0–I48.4, I48.9) [21,22].

The Latvian NHS maintains a database of all reimbursed outpatient care prescription data, including both paper and electronic prescriptions. This database captures all medicine dispensed at pharmacies and reimbursed for patients by the government. As a result, the study analysed all dispensed medicine prescriptions during the study period. In particular, the study could not collect data on prescriptions prescribed to patients but not purchased. Non-proceeded paper prescriptions remained with the patients, so these data were unavailable, but electronically prescribed prescription data were saved separately on the eHealth platform used for prescribing. The reimbursement system did not possess this information if an electronic prescription had not been used. In Latvia, the NHS specifies the list of active substances that physicians are permitted to prescribe for each diagnosis. The NHS also maintains a list of reimbursable medications based on diagnosis codes dispensed at pharmacies. If a physician prescribed an active substance for AF and flutter or if a pharmacist dispensed medication outside the approved list of reimbursable medications, it was considered a medication error.

Each prescription record contained details such as the patient’s age, gender, and specific information about the prescribed medicine, including the marketing authorisation number, brand name or international non-proprietary name, dosage, and physician’s speciality. Furthermore, the records also encompassed data regarding the dispensed medicine’s marketing authorisation number, brand name, derivative INN, dosage, pharmacy identification code, and whether the prescription was in the paper or electronic format. Patients were categorised as “switchers” if they had transitions between different anticoagulant molecules throughout their therapeutic regimen.

Ethical approval for this study was obtained from the Central Medical Ethics Committee of Latvia on 4 November 2021 (No. 10900) and carried out in accordance with the Declaration of Helsinki. In order to safeguard the privacy and security of patient information, the Latvian NHS furnished the study with anonymous and encrypted records. These measures were implemented to prevent the reverse identification of patients.

### 2.2. Measurements of Anticoagulant Therapy Adherence

This cohort included patients who had been prescribed at least two DOAC prescriptions, including rivaroxaban, dabigatran, edoxaban, or apixaban, to evaluate their adherence. Adherence to warfarin therapy was not assessed due to the unstable nature of the treatment regimen, which is based on international normalised ratio results.

Medication adherence was evaluated through a modified proportion of day covered (PDC) approach, drawing upon dispensing data retrieved from the NHS prescription database. Higher PDC values signify higher medication adherence. This technique guarantees accurate adherence evaluation whilst considering continuous treatment and adequately avoiding exaggerated estimation of adherence. Although multiple strategies exist for calculating medication adherence via pharmacy refill data, the PDC metric was chosen because of its endorsement by the Pharmacy Quality Alliance and prior evidence supporting its effectiveness across numerous studies [16,23,24,25].

To account for the possibility of patients stockpiling their medications at home, the study allowed for overlaps between prescriptions to be included in the PDC calculation. The PDC method is a widely recognised approach for evaluating medication adherence, with values ranging from 0 to 100%. Achieving adherence levels higher than 80% is crucial for preventing negative health outcomes, particularly in cases when missing even a single dose can significantly impact disease management. This is particularly true for DOACs, which have a short half-life (5–14 h) and require consistent daily dosing to maintain therapeutic anticoagulation levels [16,26]. Given that previous research has demonstrated that a PDC of 90% is associated with improved clinical outcomes in addition to the standard 80% benchmark, this value was also included in the analysis [17].

The PDC method was employed to compute medication adherence, involving the division of total days covered by the medicine regimen by the interval between the patient’s initial and final dispensing dates, omitting the last prescription. Overlapping fills for the same drug due to early refilling were accommodated by adjusting the dispensed medication quantity to include the remaining doses alongside the new supply [24,25,27]. Within this study, adherence rates were classified into three groups: (1) below 80%, (2) between 80% and 90%, and (3) above 90%. In accordance with the Latvia reimbursement system, prescriptions are limited to a maximum 3-month course.

### 2.3. Statistical Analysis

The assumption of data distribution was assessed by the Shapiro–Wilk test and inspection of the normal Q-Q plots. The assumption of homogeneity of variance was tested by Levene’s test. Welch’s ANOVA was used to test the difference in age between adherence groups (<80%, 80–90%, and >90%). The chi-square test of goodness of fit was employed to evaluate whether the distribution of gender and “switchers” deviates from a random distribution among the adherence groups. All statistical analyses were performed by using the Jamovi statistical software (https://www.jamovi.org, v.2.3.28 (accessed on 14 February 2024)). An alpha level of 0.05 was used for all the statistical analyses.

## 3. Results

A total of 1,646,648 prescriptions were extracted from the NHS reimbursement system, prescribed for 104,524 unique patients from 2012 to 2022. More detailed information about prescribed medicines is presented in Table 1. Out of the total prescriptions, 483,771 (29.4%) were paper prescriptions. It is noteworthy that the rate of paper prescriptions decreased significantly from 98.5% in 2017 to 19.4% in 2018 and further decreased to a minimal rate of 1.3% in 2022. General practitioners predominantly prescribed medications corresponding to code I48 at a frequency of 92.1%, followed by cardiologists (4.3%) and internists (1.3%). The utilisation of INNs in prescriptions remained relatively low, ranging from 0.2–0.3% between 2012 and 2017. However, an increase was observed in 2018 and 2019, with INN usage rates rising to 2.6% and 3.1%. This trend continued to accelerate and INNs accounted for 61.2% of prescriptions in 2020, increasing to 80.6% and 79.7% in 2021 and 2022, respectively. Specifically, 16.7% of patients selected a single pharmacy, whereas 27.7% visited either one or two pharmacies during the observed period. Meanwhile, other patients obtained medicines from multiple pharmacies.

Anticoagulation therapy was prescribed to a total of 93,721 patients, among whom 39,181 individuals (41.8%) were identified as “switchers”. Specifically, 32,842 (83.8%) patients were prescribed two different anticoagulants, 6324 (16.1%) received three, 608 (1.6%) were prescribed four, and 15 (<0.1%) individuals were prescribed five distinct anticoagulants throughout the analysed period. Significant changes occurred in anticoagulant prescriptions following 2018 (Figure 1). Warfarin prescriptions started declining as soon as DOACs became available in 2018. Although all DOACs kept on gaining market share, the number of dabigatran prescriptions began declining after 2020.

The active substances prescribed by doctors outside the approved list of active substances for AF and flutter accounted for less than 0.1% in all periods. The rate of medication errors as incorrectly dispensed medication at pharmacies remained at similar levels. Regardless, in the separate DOAC cohort of 704,521 prescriptions (Table 2), a medication error between the prescribed and dispensed dosages was detected in 1.1% of cases, affecting 3987 patients.

Out of all the cohorts, adherence levels to DOAC therapy were assessed for 45,933 patients. Two hundred and one patients were excluded from the evaluation of DOAC adherence due to having fewer than two prescriptions. The average age in the group was 75.0 (SD = 9.6) years. Among them, 27,355 (59.6%) were women. Total adherence levels to DOAC therapy were 69.4%, with specific values for individual agents being 70.7% for rivaroxaban, 69.4% for dabigatran, 64.5% for edoxaban, and 45.8% for apixaban. Only 44.2% of users achieved an adherence level exceeding 80%. More detailed information is shown in Table 3. If patients experienced a change from one DOAC to another, significant changes in adherence levels were observed in anticoagulation therapy (*p* < 0.001). Nevertheless, adherence levels to each active substance remained at the same levels (*p* = 0.577) among swithers from one DOAC to another.

## 4. Discussion

Long-term therapy with DOACs plays a key role in reducing the risk of blood clots in patients with AF. The utilisation of DOACs saw a dynamic evolution globally from 2011 onwards, immediately after market authorisation [28,29]. Rivaroxaban was the most widely used DOAC until 2018–2019, after which apixaban surpassed it to become the predominant choice among clinicians [19,30,31,32,33,34,35]. Despite global trends, it is important to note that the specific data and trends can vary by country. Specifically, our findings highlight that the use of apixaban is markedly less prevalent in Latvia. Notably, the incidence of patients transitioning between different DOACs was relatively low, indicating a tendency for patients to maintain their initial DOAC therapy. This excludes patients switching from warfarin to a DOAC, which represents a distinct category of therapeutic transition. This pattern suggests a degree of stability and satisfaction with initial DOAC regimens and the influence of clinical guideline recommendations [2].

High adherence to DOAC therapy is crucial for maintaining its therapeutic benefits. Discontinuation or inconsistent dosing can significantly reduce the effectiveness of DOACs in preventing thrombosis. In this research, over 50% of DOAC long-term users did not achieve optimal adherence levels, a higher rate than in other studies [15,16,36,37,38,39], despite one study presenting similar results [40]. Contrary to expectations, adherence to apixaban was the lowest, a finding that may be attributed to the smaller number of patients. Nevertheless, the same situation was observed in the study by Shiga et al. [17]. The limited popularity of apixaban may result from physicians’ lack of familiarity or experience with the medication, as well as the necessity of a twice-daily dosing schedule. The study’s results indicate that there was no significant association between adherence rates to DOACs and factors such as age, gender, and anticoagulant “switcher” status [22]. The association between adherence level and “switcher” status was noted but did not prove causality. This suggests that a broader range of factors contributing to non-adherence should be explored, including patient beliefs and attitudes towards medication, levels of health literacy, complex medication regimens, psychological factors, the patient–provider relationship, and others. Understanding and addressing these reasons for non-adherence are important for improving patient outcomes and ensuring the effectiveness of DOAC therapy in managing AF.

Medication errors related to prescriptions encompass a range of issues that can occur at any stage of the medication use process, from prescribing to dispensing. The study revealed that some prescribed active substances were not consistent with the approved reimbursement medication list and medical errors at dispensing, although this was infrequent. The analysis revealed inconsistencies between prescribed and dispensed doses of DOACs, affecting slightly more than 8% of all patients who were prescribed a DOAC at least once. Discrepancies in the dosage of DOACs can have significant clinical implications. Overdosing can lead to an increased risk of major bleeding events, including intracerebral haemorrhage, which is associated with increased mortality. On the other hand, underdosing may compromise the drug’s efficacy, raising the risk of systemic embolism or ischemic stroke [31,40,41]. A pragmatic challenge in anticoagulant therapy involves pharmacists dispensing alternative doses of DOACs due to constraints in market availability. While necessary, this adaptation directly influences the safety and efficacy of anticoagulant therapy. Practical software tools and the implementation of artificial intelligence that support anticoagulation can help reduce stroke risk in patients with atrial fibrillation [42]. Studies highlight the potential of artificial intelligence applications in ambulatory care settings as valuable tools to aid healthcare providers in drug management. These technologies can improve medication adherence, track patient outcomes, and optimize treatment strategies, improving patient care and outcomes [43,44].

It has been proven that providing recommendations is insufficient to bring about substantial changes in behaviour or practices. Regulation changes are necessary [45]. This study’s results also indicate a significant impact on practice by the mandatory use of electronic prescribing systems from 2018 and INN from 2020 for reimbursement purposes [46,47]. Our strong recommendation is to utilise more administrative data to assess and identify non-adherent patients at high risk. Adherence level, as an indicator, can help clinicians better estimate therapy patterns that describe patient behaviour and medication-taking habits. Integrating medication adherence assessment as an additional parameter into the electronic prescription database presents an excellent opportunity to enhance patient care and outcomes.

The study is subject to certain limitations. It is important to note that data availability within the system limited the study, due to the omission of prescribed but not dispensed prescription data from the system, including patients’ hospitalisations and the use of medicine there, which remained unanalysed, thus potentially impacting the study outcomes. Furthermore, the study relied on quantitative adherence measures derived from medication prescription refill data, neglecting to delve into patient behaviours and factors influencing non-adherence. Simply refilling a prescription does not guarantee adherence. Important qualitative elements such as patients’ socio-demographic details, disease duration, symptom severity, comorbidities, and health literacy were not explored, limiting the depth of analysis regarding non-adherence. Lastly, while quantitative metrics offer insights into medication accessibility, they do not directly confirm ingestion or adherence.

## 5. Conclusions

Of all the DOACs, rivaroxaban had the highest utilisation rate for AF and flutter in Latvia, while apixaban was unpopular, and the usage of dabigatran had been declining. The total adherence level to DOAC therapy is evaluated as low because less than half of the patients achieved optimal values and there was no significant difference in age, gender, or “switcher” status among adherence groups. The total adherence level to DOAC therapy is evaluated as low because less than half of the patients achieved optimal values and there was no significant difference in age, gender, or “switcher” status among adherence groups. More patient-focused research is necessary to understand the reasons for non-adherence. Physicians’ prescribing have changed over a decade, influenced by regulatory changes such as the mandatory use of INN and electronic prescription systems. The risk of dispensing an incorrect dose at the pharmacy persists. More advanced electronic control systems are required to minimise medication prescribing and dispensing errors.

## Figures and Tables

**Figure 1 healthcare-12-00975-f001:**
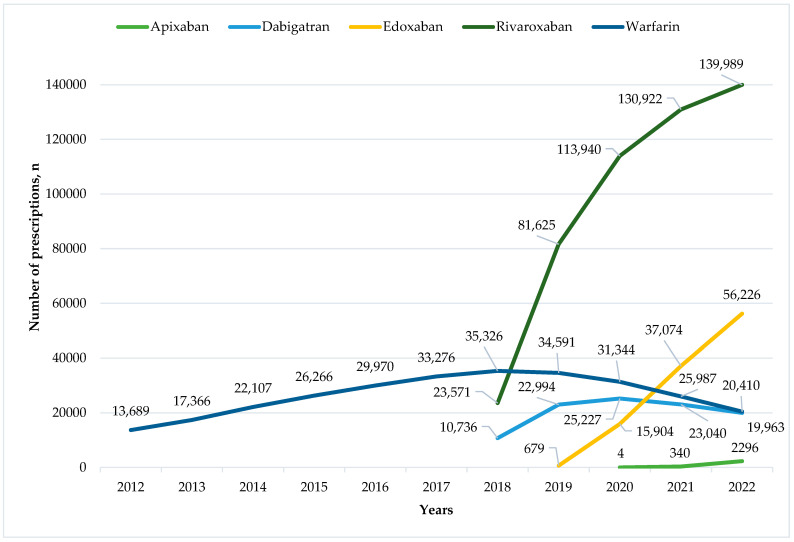
The trend of anticoagulant prescribing for atrial fibrillation and flutter between 2012 and 2022.

**Table 1 healthcare-12-00975-t001:** Overview of prescribed active substances and the number of prescriptions for atrial fibrillation and flutter.

Active Substance	2012	2013	2014	2015	2016	2017	2018	2019	2020	2021	2022
	Number of Prescriptions, n (%)										
Aethacizine	7322	9369	11,150	14,132	16,688	19,006	22,515	26,852	30,007	31,203	33,518
	(14.5)	(15.6)	(15.9)	(17.6)	(18.8)	(20.0)	(16.6)	(12.7)	(11.6)	(10.9)	(10.8)
Amiodarone	16,521	15,922	15,457	15,298	14,839	14,388	14,575	14,781	12,741	10,969	10,319
	(32.7)	(26.4)	(22.1)	(19.1)	(16.7)	(15.1)	(10.7)	(7.0)	(4.9)	(3.8)	(3.3)
Apixaban	-	-	-	-	-	-	-	-	4	340	2296
									(0.0)	(0.1)	(0.7)
Atenolol	40	29	43	28	64	47	26	45	32	14	-
	(0.1)	(0.0)	(0.1)	(0.0)	(0.1)	(0.0)	(0.0)	(0.0)	(0.0)	(0.0)	
Dabigatran	-	-	-	-	-	-	10,736	22,994	25,227	23,040	19,963
							(7.9)	(10.8)	(9.8)	(8.0)	(6.5)
Digoxin	2933	3175	3493	3880	4123	4497	4705	4886	5214	5349	5177
	(5.8)	(5.3)	(5.0)	(4.8)	(4.6)	(4.7)	(3.5)	(2.3)	(2.0)	(1.9)	(1.7)
Diltiazem	191	173	148	116	115	109	85	105	64	35	37
	(0.4)	(0.3)	(0.2)	(0.1)	(0.1)	(0.1)	(0.1)	(0.0)	(0.0)	(0.0)	(0.0)
Edoxaban	-	-	-	-	-	-	-	679	15,904	37,074	56,226
								(0.3)	(6.2)	(12.9)	(18.2)
Clopidogrel	354	1958	3688	5025	6083	6577	6179	5561	4733	3895	3274
	(0.7)	(3.2)	(5.3)	(6.3)	(6.9)	(6.9)	(4.5)	(2.6)	(1.8)	(1.4)	(1.1)
Clopidogrel/Aspirin	57	1244	2142	2702	2738	2320	2089	1812	1432	1150	773
	(0.1)	(2.1)	(3.1)	(3.4)	(3.1)	(2.4)	(1.5)	(0.9)	(0.6)	(0.4)	(0.3)
Metoprolol	5110	5861	5843	5823	6144	6335	6248	6975	6275	5857	5700
	(10.1)	(9.7)	(8.4)	(7.3)	(6.9)	(6.7)	(4.6)	(3.3)	(2.4)	(2.0)	(1.8)
Propafenone	3765	4624	5349	6440	7423	8128	9350	10,810	10,929	10,751	11,054
	(7.4)	(7.7)	(7.6)	(8.0)	(8.4)	(8.5)	(6.9)	(5.1)	(4.2)	(3.7)	(3.6)
Rivaroxaban	-	-	-	-	-	-	23,571	81,625	113,940	130,922	139,989
							(17.3)	(38.5)	(44.1)	(45.6)	(45.3)
Warfarin	13,689	17,366	22,107	26,266	29,970	33,276	35,326	34,591	31,344	25,987	20,410
	(27.1)	(28.8)	(31.6)	(32.7)	(33.8)	(35.0)	(26.0)	(16.3)	(12.1)	(9.1)	(6.6)
Verapamil	600	522	500	485	464	486	470	274	241	268	253
	(1.2)	(0.9)	(0.7)	(0.6)	(0.5)	(0.5)	(0.3)	(0.1)	(0.1)	(0.1)	(0.1)
Others *	11	6	3	29	18	5	7	1	4	4	5
	(0.0)	(0.0)	(0.0)	(0.0)	(0.0)	(0.0)	(0.0)	(0.0)	(0.0)	(0.0)	(0.0)
Total	50,593	60,249	69,923	80,224	88,669	95,174	135,882	211,991	258,091	286,858	308,994

* Active substances prescribed outside the approved reimbursement list for AF and flutter (I48.0–I48.4, I48.9).

**Table 2 healthcare-12-00975-t002:** Conformity of physicians’ prescribed and dispensed medicine in the pharmacy.

Prescribed Medication by a Physician		Dispensed Medication at a Pharmacy							
Active Substance	Dosage	Apixaban		Dabigatran		Edoxaban		Rivaroxaban	
		2.5 mg	5 mg	110 mg	150 mg	30 mg	60 mg	15 mg	20 mg
Apixaban	2.5 mg	1215	16						
	5 mg	10	1399						
Dabigatran	110 mg			50,334	595			1	
	150 mg			740	50,242				
	75 mg			1	1				
Edoxaban	15 mg					14	1	3	
	30 mg				1	53,223	472	3	2
	60 mg					624	55,490	2	1
Rivaroxaban	10 mg							19	22
	15 mg				3			211,346	2127
	20 mg							2965	273,455

**Table 3 healthcare-12-00975-t003:** Characteristics of patients for whom adherence level was assessed (n = 45,933).

Adherence Group	Number of Patients, n	Age, Years (±SD)	Female Gender, n (%)	Switchers from Warfarin to DOAC, n (%)	Switchers among DOACs, n (%)
<80%	25,615	75.1 (9.7)	15,116 (59.0)	7328 (28.6)	3515 (13.7)
80–90%	10,483	75.0 (9.4)	6434 (61.4)	3071 (29.3)	1034 (9.9)
>90%	9835	74.8 (9.8)	5805 (59.0)	2583 (26.3)	458 (4.7)
		*p =* 0.911	*p =* 0.978	*p =* 0.898	*p =* 0.122

## Data Availability

Data are contained within the article.

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
