# Peer review of "Role of a National Health Service Electronic Prescriptions Database in the Detection of Prescribing and Dispensing Issues and Adherence Evaluation of Direct Oral Anticoagulants"

_healthcare, 2024, doi:10.3390/healthcare12100975_

Round 1

Reviewer 1 Report

Comments and Suggestions for Authors

Dear authors,

Firstly, I would like to express my gratitude to the journal for providing me with the opportunity to review the manuscript titled "Role of National Health Service Electronic Prescriptions Database in Detection of Prescribing and Dispensing Issues and Adherence Evaluation of Direct Oral Anticoagulants". Anticoagulants are among the most vital medications, and their prescription in certain countries can significantly impact healthcare expenditure. Therefore, the monitoring of their usage is a topic of current interest. Congratulations on identifying this and addressing it in your manuscript.

Although the article is interesting, it needs improvement before it can be considered for publication:

Abstract: The presentation of results needs improvement, as this issue recurs throughout the manuscript. Some parts are difficult to read and overly verbose.

Introduction: The objective of the study needs better clarification because the reason for conducting it is not clear.

Materials and Methods: They are overly verbose and need to be rewritten more clearly. Additionally, the statistical analysis used should be specified better.

Discussion:

The concept of medication errors needs to be explained better, particularly concerning anticoagulants. I recommend considering some articles that have delved into this topic (10.1161/STROKEAHA.116.015468 and 10.1136/bmjopen-2022-065301).

Overall, while the manuscript shows promise, these areas need to be addressed to enhance its quality and suitability for publication.

Comments on the Quality of English Language

A Moderate editing of English language is required

Author Response

Dear reviewer,

Thank you for taking the time to read and provide feedback on the study.

Abstract: The presentation of results needs improvement, as this issue recurs throughout the manuscript. Some parts are difficult to read and overly verbose.

We apologize if there was any misunderstanding. We acknowledge the need for improvement in presenting the results, but the data in the reimbursement system is limited. Please note that the abstract has a word limit, which restricts the level of detail we can provide. Please see the changes in the manuscript.

Introduction: The objective of the study needs better clarification because the reason for conducting it is not clear.

The aim of this study was to conduct an analysis of prescribing and dispensing patterns, including evaluation of DOAC therapy adherence, among patients with AF based on data from the National Health Service's reimbursement prescriptions database. Based on this purpose, we described statistics, DOAC's importance in clinical practice, non-adherence problems and how administrative data can be used to analyse actual practice at the state level. We have added the definition of medication error to the Introduction. We desired to emphasize the main topics without adding extra text that might disrupt the understanding of the core idea.

Materials and Methods: They are overly verbose and need to be rewritten more clearly. Additionally, the statistical analysis used should be specified better.

Thank you for your feedback on the Materials and Methods section. We have provided a detailed methodology to ensure that others can replicate our methods effectively. The background information on local regulation and the reimbursement system was included to clarify study details. We revised the section to enhance clarity.

The concept of medication errors needs to be explained better, particularly concerning anticoagulants. I recommend considering some articles that have delved into this topic (10.1161/STROKEAHA.116.015468 and 10.1136/bmjopen-2022-065301).

Thank you for your comment on medication errors. We highly appreciate your recommendation and have integrated related revisions into the Discussion section. Additionally, we have considered the articles you recommended to enhance the explanation and understanding of this topic.

Thank you once again for your valuable input in enhancing the quality of this manuscript.

Kind regards,

Anna Gavrilova

Reviewer 2 Report

Comments and Suggestions for Authors

Solid paper using Real World Data in adressing adherence issues with DOACs.

A few punctual remarks on the manuscript:

Line 21: Remove ICD-10 codes from the abstract

Line 30: Remove “and dispensing”, because physicians do not dispense medications

Line 45: Replace “Recently” by “over the last decade”

Line 78: the NHS database captures outpatient reimbursement. What if a patient was hospitalized during the study period? Will he continue to use his home medication, or does the hospital provide (uncaptured) medicines during hospital stay, and thus underestimating PDC ? If this is the case, this should be addressed, for instance after Line 242 discussing limitations.

Line 93-94: Replace “ anticoagulants” by “DOAC molecules”

Line 101-102 : Specify “at least two DOAC medications”. Is this at least 2 boxes, at least 2 prescriptions, at least 2 molecules ?

Line 152/153 (and also in abstract) : 16.7% of patients always went to the same pharmacy, and 27.7% used 2 or more pharmacies. What about the 55% other patients ? Where did they get their DOAC dispensed ?

Line 160-162 : the authors should state that “warfarine prescriptions started declining as soon as DOACs became available” and that “all DOACS kept on gaining market share, except dabigatran”

Figure 1 : add the indications in the Figure caption (the same way as in th Table 1 caption)

Table 3 : header of the 4th colums shoud read “Female gender”

Line 189/190 : the data shown does not allow to state this evolution from 2014 onwards. The authors should specify at what moment DOACs were approved for AF and flutter (be it via the marketing authorisation, or via the Latvian reimbursement list)

Comments on the Quality of English Language

English style is fine

Author Response

Dear reviewer,

Thank you for taking the time to review our study named Role of National Health Service Electronic Prescriptions Database in Detection of Prescribing and Dispensing Issues and Adherence Evaluation of Direct Oral Anticoagulants. We appreciate your feedback and suggestions for improvement.

A few punctual remarks on the manuscript:

Line 21: Remove ICD-10 codes from the abstract

Line 30: Remove “and dispensing”, because physicians do not dispense medications

Line 45: Replace “Recently” by “over the last decade”

We have made these changes.

Line 78: the NHS database captures outpatient reimbursement. What if a patient was hospitalized during the study period? Will he continue to use his home medication, or does the hospital provide (uncaptured) medicines during hospital stay, and thus underestimating PDC ? If this is the case, this should be addressed, for instance after Line 242 discussing limitations.

We appreciate your concern about hospitalization affecting PDC adherence rates. When hospitalization is scheduled, patients typically use home medications; however, hospitals usually provide all necessary medicines. PDC accounts for this, classifying patients as adherent only when PDC >80%. We agree with your suggestion to include uncaptured medications used during hospital stays in the limitations section.

Line 93-94: Replace “ anticoagulants” by “DOAC molecules”

Thank you, we changed it.

Line 101-102 : Specify “at least two DOAC medications”. Is this at least 2 boxes, at least 2 prescriptions, at least 2 molecules ?

Thank you for pointing that out. We've corrected it to "prescriptions" as intended.

Line 152/153 (and also in abstract) : 16.7% of patients always went to the same pharmacy, and 27.7% used 2 or more pharmacies. What about the 55% other patients ? Where did they get their DOAC dispensed ?

We have included additional details to explain patients' selection of a pharmacy for their medication. This text has also been added to the abstract.

“Specifically, 16.7% of patients selected a single pharmacy, whereas 27.7% visited either one or two pharmacies during all observed periods. Meanwhile, other patients obtained medicines from multiple pharmacies.

Line 160-162 : the authors should state that “warfarine prescriptions started declining as soon as DOACs became available” and that “all DOACS kept on gaining market share, except dabigatran”

We made changes to these sentences.

Figure 1 : add the indications in the Figure caption (the same way as in th Table 1 caption)

Table 3 : header of the 4th colums shoud read “Female gender”

We appreciate your attention to detail and have updated Figure 1 and Table 3 accordingly.

Line 189/190 : the data shown does not allow to state this evolution from 2014 onwards. The authors should specify at what moment DOACs were approved for AF and flutter (be it via the marketing authorisation, or via the Latvian reimbursement list)

We mentioned 2014 instead of 2011 based on the spread of rivaroxaban in clinical practice. After thoroughly reviewing additional literature and marketing authorization information, we have incorporated more specific details and relevant references. Please see the changes in the manuscript.

Thank you one more time for your contribution to improving this manuscript's quality.

Kind regards,

Anna Gavrilova

Reviewer 3 Report

Comments and Suggestions for Authors

This study aimed to conduct an analysis of prescribing and dispensing patterns, including evaluation of DOAC therapy adherence among patients with atrial fibrillation, based on data from the National Health Service's (NHS) reimbursement prescriptions database. Given the pharmacotherapeutic profile of this class of drugs, the analysis is pertinent. Overall, the background focuses on the most relevant aspects, and the materials and methods are adequately described. However, the presentation of the results needs improvement.

In Table 1, the authors need to specify that aspirin is low-dose and explain what the "others" refer to. Figure 1 should be transformed into a table to allow for a more detailed analysis of the results regarding anticoagulant prescriptions. This table should include a note regarding the year each DOAC was authorized for market introduction, as we are comparing drugs with different market times. Additionally, the authors state in the results: "Active substances prescribed were inconsistent with the approved reimbursement medication list, and the issue of incorrectly dispensed active substances against prescription at a pharmacy was lower than 0.1% in all periods. However, in the DOAC prescriptions group, there were 1.1% inconsistencies out of 704,521 prescriptions between the prescribed and dispensed dosages, which impacted 3,987 patients." However, these data need to be clearly presented. Interpretation of Table 2 can lead to interpretation errors and therefore needs improvement. In Table 3, the authors should specify the gender to which the values presented refer.

Comments on the Quality of English Language

Moderate editing of English language required

Author Response

Dear reviewer,

Thank you for examining our study named Role of National Health Service Electronic Prescriptions Database in Detection of Prescribing and Dispensing Issues and Adherence Evaluation of Direct Oral Anticoagulants. We value your recommendations for manuscript enhancement.

In Table 1, the authors need to specify that aspirin is low-dose and explain what the "others" refer to.

We apologize if there was any misunderstanding. Now, we specified that low-dose aspirin was mentioned in Table 1. And we definitely agree with you that an explanation of Others* should be added. Please review the updates made to the manuscript.

Figure 1 should be transformed into a table to allow for a more detailed analysis of the results regarding anticoagulant prescriptions. This table should include a note regarding the year each DOAC was authorized for market introduction, as we are comparing drugs with different market times.

Thank you for your comment. Please note that the same information is present in Table 1, showing that rivaroxaban and dabigatran were included in the Latvian reimbursement system in 2018, edoxaban in 2019 and apixaban in 2020, respectively. DOAC centralised market authorisation in Europe was earlier, starting in 2011. In this study, we wanted to emphasise the importance of prescription data among long-term patients with AF and flatter, so we selected data from the reimbursement system, not electronic prescription platforms. We disagree with converting Figure 1 into a table, as we believe that visualising the data, at least partially, is essential. We hope for your understanding.

 Additionally, the authors state in the results: "Active substances prescribed were inconsistent with the approved reimbursement medication list, and the issue of incorrectly dispensed active substances against prescription at a pharmacy was lower than 0.1% in all periods. However, in the DOAC prescriptions group, there were 1.1% inconsistencies out of 704,521 prescriptions between the prescribed and dispensed dosages, which impacted 3,987 patients." However, these data need to be clearly presented. Interpretation of Table 2 can lead to interpretation errors and therefore needs improvement.

Following your advice, we have revised the paragraph on medication errors to specify the type of inconsistency. Table 2 highlights the issue that patients do not consistently receive the prescribed dose from their doctors. Incorrect dosages can lead to a higher risk of adverse events, such as bleeding, or a suboptimal therapeutic effect, increasing the risk of thrombosis. Furthermore, we have added a more detailed information of medication errors to section 2.1. Study Subjects and Data Collection. We hope this additional information will help clarify the types of errors we calculated for this study.

In Table 3, the authors should specify the gender to which the values presented refer.

Thank you for your attention to detail. We added that it was the female gender.

Thank you once again for your valuable input in enhancing this manuscript.

Kind regards,

Anna Gavrilova

Round 2

Reviewer 1 Report

Comments and Suggestions for Authors

The changes you have made to the manuscript are very clear. Congratulations, now the article is suitable for publication!

Reviewer 3 Report

Comments and Suggestions for Authors

Regarding this article, the authors took into consideration the comments made and made alterations accordingly. There was only one suggestion that the authors did not accept, but they justified it, and the response to me is sufficient. The article was clearly improved and therefore can be published.

Comments on the Quality of English Language

Minor editing of English language required